# Reprograming of Gene Expression of Key Inflammatory Signaling Pathways in Human Peripheral Blood Mononuclear Cells by Soybean Lectin and Resveratrol

**DOI:** 10.3390/ijms232112946

**Published:** 2022-10-26

**Authors:** Nilofer Qureshi, Julia Desousa, Adeela Z. Siddiqui, David C. Morrison, Asaf A. Qureshi

**Affiliations:** 1Department of Biomedical Sciences, Shock/Trauma Research Center, School of Medicine, University of Missouri, 2411 Holmes Street, Kansas City, MO 64108, USA; 2Pharmacology/Toxicology, School of Pharmacy, University of Missouri-Kansas City, Kansas City, MO 64108, USA

**Keywords:** LPS, signal transduction, resveratrol, lectins, cytokines, NO, RNAseq

## Abstract

Inflammation is linked to several human diseases like microbial infections, cancer, heart disease, asthma, diabetes, and neurological disorders. We have shown that the prototype inflammatory agonist LPS modulates the activity of Ubiquitin-Proteasome System (UPS) and regulates transcription factors such as NF-κB, leading to inflammation, tolerance, hypoxia, autophagy, and apoptosis of cells. We hypothesized that proteasome modulators resveratrol and soybean lectin would alter the gene expression of mediators involved in inflammation-induced signaling pathways, when administered ex vivo to human peripheral blood mononuclear blood cells (PBMCs) obtained from normal healthy controls. To test this hypothesis, analysis of RNA derived from LPS-treated human PBMCs, with or without resveratrol and soybean lectin, was carried out using Next Generation Sequencing (NGS). Collectively, the findings described herein suggest that proteasome modulators, resveratrol (proteasome inhibitor) and lectins (proteasome activator), have a profound capacity to modulate *cytokine* expression in response to proteasome modulators, as well as expression of mediators in multiple signaling pathways in PBMCs of control subjects. We show for the first-time that resveratrol downregulates expression of mediators involved in several key signaling pathways *IFN-γ*, *IL-4*, *PSMB8* (LMP7), and a subset of LPS-induced genes, while lectins induced *IFN-γ*, *IL-4*, *PSMB8*, and many of the same genes as LPS that are important for innate and adaptive immunity. These findings suggest that inflammation may be influenced by common dietary components and this knowledge may be used to prevent or reverse inflammation-based diseases.

## 1. Introduction

Inflammation is a major contributing factor in the outcomes of diseases like infections, cancer, heart disease, asthma, diabetes, and neurological disorders. Bacterial components such as toxic LPS, peptidoglycan and CpG DNA elicit a cytokine storm; inflammation leading to hypoxia, oxygen deprivation in the tissues and cells, and, finally, cell death. This may lead to septic shock, particularly in individuals with underlying diseases [1,2]. COVID-19 viral proteins are also known to elicit a cytokine storm; the response to COVID-19 is highly variable, though, and can range from minor or major depending on the robustness of the host’s immune system. Such mechanisms are like those occurring during microbial sepsis and septic shock [3,4,5,6,7,8,9]. Unfortunately, there is currently no effective treatment available for diseases such as sepsis, with supportive therapy or antibiotic treatment.

Our laboratory focuses on defining the basic underlying molecular and cellular mechanisms by which microbial proinflammatory mediators such as LPS function to regulate inflammation [10,11,12,13,14,15,16,17]. Our published and preliminary data have helped to establish that distinct phases of inflammation or tolerance can be defined, based on the enzymatic status of the host inflammatory cell proteasomes as mechanisms reproducibly dictating LPS action [13,18,19,20,21]. Although all cells contain proteasomes, they differ in their subunit composition [14]. It has long been recognized that the Ubiquitination-Proteasome System (UPS) plays a major role in intracellular protein degradation of short-lived proteins in a cell. UPS also affects the level of gene expression of multiple genes via, degradation/activation of transcription factors and the process of inflammation. Proteasomes are known to be present in the cell membranes, cytoplasm, and the nucleus [10,13]. They are potentially an attractive therapeutic target for maintaining a strong immune system and for reprogramming the cells’ selective processes ranging from proliferation, differentiation, and under some circumstances cell death. 

The cell’s proteasome subunit composition and potential changes to that composition control its enzymatic activities lead to regulation of degradation of crucial Toll-like receptors (TLR)-activated signaling proteins and key metabolic proteins. For example, the 20S proteasome contains at least three major protease activities- specifically chymotrypsin-like (X, LMP7), post-acidic (Y, LMP2), and the trypsin-like activities (Z, LMP10) [18]. After activation of the proteasome with proinflammatory mediator LPS, the constitutive form expressing predominantly X, Y, Z subunits, are replaced with subunits low molecular weight subunits LMP7, LMP2, and LMP10 (associated with catabolic functions), respectively, in newly synthesized termed immunoproteasomes [10,11,12,13,14,15,16,17]. Originally, immunoproteasomes were known for their capacity to degrade proteins into small peptides for antigen presentation, and to rid the cell of any misfolded proteins. More recently, however, we have established that proteasome inhibitors that inhibit protein degradation by the UPS affect regulation of multiple signaling pathways activated by proinflammatory mediators and agonists [12,22]. We have also shown that when subunits of the immunoproteasome cannot be upregulated any further leads to the development of a state of tolerance, and the cells become refractory to agonists [14, unpublished data]. Thus the composition of subunits and levels of proteasome’s proteases ultimately dictate the regulatory functions in various cells [10,11,12,13,14,15,16,17]. Most of these studies were carried out with mouse macrophages. The exact mechanisms by which LMP subunits regulate cellular signaling are however, currently not known. 

Cells of the hematopoietic system (more differentiated cells) are known to contain more immunoproteasome subunits (LMP7. LMP2), as compared to tissue cells or some mouse cells which contain primarily (XYZ) constitutive type subunits [14]. LPS for example, can function to differentially modulate gene expression of certain cytokines/proteins in cells. We have identified resveratrol (found in berries, red wine, and other plant sources) as an inhibitor of LMP7 protease which inhibits degradation of tryptophan containing peptides more effectively and to a lesser extent X (tyrosine, proteasome subunits that have the CT-like activity) in human monocytes [14,15,16,17]. We have, on the other hand, also identified soybean lectin as an activator of the LMP7. The exact mechanisms by which these agents function in PBMCs to modulate the gene expression of proteins involved in LPS-induced (w/wo) signaling pathways has not been determined. 

Collectively, our published and preliminary data and those of others suggest that a differential expression of macrophage (innate immunity) IL-12, IL-1β, IL-6, IL-12 and TNF-α, and T cell cytokines (acquired immunity), such as IFN-γ, and IL-4, which are modulated during development of inflammation and tolerance [12,13,15]. IFN-γ is known to regulate cell differentiation, cell growth, and level of expression of many key genes to the microbicidal development of host response to microbes by upregulation of LMP7 subunit (Th1 response) [14,15,16,17,22,23,24,25,26,27,28]. However, IL-4 is also induced, mainly by the host Th2 cells and this cytokine can counteract the effects of IFN-γ on Mφ by inducing new expression of both XYZ and LMP subunits leading to growth of new cells [29,30,31,32,33]. These findings strongly implicate the existence of a highly regulated coordinated control of immune activity by changing proteasome’s subunit activity (leading to differential degradation of proteins, transcription factors, and enzymes) and activation of alternate pathways involved in innate and acquired immunity. 

Our overall hypothesis is that the changes in level of gene expression triggered by LPS and/or other proinflammatory stimuli in PBMCs from healthy individuals will be differentially affected by plant constituents such as resveratrol (RES, inhibitor of LMP7) and soybean lectins (LEC, activator of LMP7) found in human diets [34,35], which were associated with well-characterized alterations in gene expression of mediators of multiple signaling pathways, the UPS, and associated proteasome’s subunit expression levels. LPS (from bad gut bacteria) and LEC (in plants lectins and soybeans), and RES (in fruits and vegetables) may be altering signaling pathways (inflammation) in human PBMCs. In this study, LPS and LEC both upregulated gene expression of proinflammatory responses (LMP7, IFN-γ and IL-4) and showed some synergism, while RES downregulated these proinflammatory responses. 

## 2. Results

Figure 1A–C LPS and LEC50 upregulates expression of 698 genes, and RES showed reduced expression of inflammatory genes using next generation sequencing (NGS). LPS and LEC50 upregulated expression of 698 genes, and RES showed reduced expression of several genes (Figure 1A–C). RES downregulated some LPS-induced genes in all pathways except for Gαs, cAMP-mediating, and glioblastoma multiforme signaling pathways (Figure 2). Gene expression modulated by RES alone in human PBMCs: RES inhibits gene expression of proteasome’s subunits, compared to controls, (Figure 3). RES primarily acts as a LMP7 immunoproteasome inhibitor (by blocking the CT-like activities of the proteasome and the gene expression of PSMB8; LMP7), however, it also robustly inhibits gene expression of several interferon-γ inducible genes Table 1 and Appendix A. This result was not observed previously with lactacystin, another proteasome inhibitor specific for X subunit (CT-like activity of the proteasome) [12].

Resveratrol downregulates some of the inflammation-linked genes in absence of LPS: Resveratrol is present as a dietary component in several fruits and has been shown to be very important for heart health. To analyze the RES-modulated genes, we have included genes that were either downregulated or upregulated by RES, as compared to control PBMCs in Table 1 and Appendix A. A total of 20 of the most significantly downregulated inflammation related genes, were chemokine genes (CCL2, MCP1, CCL7, like MCP1, CCL8 and MCP2); thrombospondin 1 (THB1); thrombomodulin (THBD); cytochrome P450 family (CYP181, CYP181-AS1); growth factor genes (EGR1 and 2); RAS protein activator like1 (RASAL1); G protein coupled receptor 68 (GPR68); and prostaglandin F2 receptor inhibitor (PTGFRN). Other significant genes included fucosidase alpha1L1 tissue (FUCA1), peroxisomal biogenesis factor 11 alpha (PEX11A), lymphocyte antigen 6 (LY6K), T-cell receptor alpha variable 19 (TRAV19), integrin beta 8 (ITGB8), interleukin 10 (IL-10), and sialidase 4 (NEU4). Gene expression of select cytokines was downregulated when RES alone was used. In contrast, some genes listed in Appendix A were upregulated by RES, and some of these encode integrin alpha D (ITGAD-CTC), Nuclear Paraspeckle Assembly Transcript 1 (VINC, virus inducible noncoding RNA) and Transmembrane 119 (HGNC 27884, Riken cDNA). 

Resveratrol downregulates certain crucial LPS-induced genes such as IFN-γ linked, early growth response 1 and 2 (*EGR1* and *EGR2*), colony stimulating factor 2 (*CSF2*), and chemokines involved in inflammation: Gene expression modulated by RES and LPS in human PBMCs. A total of 680 genes were identified as being significantly modulated by various agonists such as LPS (Figure 1). When PBMCs were treated concurrently with RES and LPS, gene expression of only 424 genes was modulated, of which 290 were downregulated. Analysis by Ingenuity Pathways Analysis (IPA) of the RNAseq data obtained from PBMCs revealed that genes involved in all major signaling pathways, were activated by LPS (some shown in Figure 2), except for *PPARγ* (nuclear receptor involved in fatty acid storage, and glucose metabolism), the retinoid receptor (*LXR/XRX*), and antioxidant activity of vitamin C signaling pathways, which were robustly downregulated.

Genes that were modulated by RES are listed in Table 2, respectively. Downregulated genes include those that encode for cytokines such as interleukin 19 (*IL-19*), interferon-γ (*IFNG*), interleukin 10 (*IL-10*), interferon-induced protein 44L (*IFI44L*), and other interferon-induced proteins, *IL-27*, *IL-36G*, chemokines, such as *CCL2*, *CCL8*, *CCL7*, and *CXCL 9*, *10* and *11*; enzymes (genes listed in Table 2 and several others are reported in Appendix A such as oligoadenylate synthase 1, 2 and 3 (*OASI 1*,*2*,*3*), gamma-glutaminyl transferase 5 (*GGT5*), sialidase, (*NEU4*), helicase with zinc finger 2 transcription coactivator (*HELZ2*), ubiquitin specific peptidase 18 (*USP18*), maltase-glucoamylase (*MGAM*); and growth factors such as colony stimulating factor 2 (*CSF2*), early growth response 2 (*EGR2*), and platelet-derived growth factor beta polypeptide, (*PDGF*).

Gene expression of proteins modulated (upregulated/downregulated) by LPS and LEC (alone or in combination) in human PBMCs: To investigate the modulation of gene expression by LEC10, we compared it with LPS, a potent bacterial agonist (Table 3A–C and more in Appendix A). Surprisingly, the analysis of these data revealed that LEC10 and LEC50 upregulate expression of many of the same genes as LPS, as compared to the vehicle control (Table 3A and a complete list in Table 3).

A few genes, however, were not upregulated by LEC10 alone to the same extent as LPS, and those are listed in Table 3B. These genes include *IFNG*, ubiquitin specific peptidase 18 (*USP18*), chemokine (C-X-C) ligand 11 (*CXCL11*), radical S-adenosyl methionine domain containing 2 (*RSAD2*), interferon alpha-inducible protein 6 (*IFI6*), calcium channel voltage dependent P/Q type alpha 1A (*CACNA1A*), family with sequence similarity 19 (*FAM19A2*), lysosomal-associated membrane protein 3 (*LAMP3*), BCL-like 14 (*BCL2L14*, *apoptosis facilitator*), Sialic acid binding Ig-like lectin (*SIGLEC1*), ubiquitin-like modifier (*ISG15*), and Cyclin A1 (*CCNA1*) (Table 3B). Genes that are upregulated by LPS, LEC10, LEC10 + LPS, LEC50 and LEC50 + LPS are listed in Table 3C, and in Appendix A.

LPS-induced and LEC-induced gene expression of important cytokines and crucial mediators of inflammation and acquired immunity, as compared to controls. Some of these included *IFN-**γ*, interferon λ *(IFN-*λ*)*, *IL-12A*, *CD80*, *CTLA*, checkpoint inhibitor in cancer cells, *STAT1*, *2*, *HLA-DQB*, *MHC II*, *IL-27*, Nuclear receptor coactivator 7 *NCOA7*, enhances the transcriptional activities of several nuclear receptors; *PTGER3*, *IP10*, *IL-8*, and others as compared to control PBMCs. Some of the upregulated genes included *TLR9* (receptor for CpG DNA), and *CLEC6A* (receptor for lectins). The healthy control PBMC, robustly induced gene expression of *IFN-**γ*, *TNF-α*, *LMP7*, nitric oxide synthase (*iNOS*, important for induction of NO), and signaling via transcription factor STAT1, and several other mediators in the presence of LPS and LEC. No upregulation of LMP10 was observed in PBMC. However, Insulin receptor was downregulated with LPS and LEC in normal controls. (Table 4). LPS and LEC upregulated MHC Class II gene expression, whereas RES downregulated their expression. Most of the genes important for canonical pathways were slightly downregulated, but those associated with IFN-γ, (Th1 response), STAT1, and pathogen-associated molecular pattern signaling pathways were strongly downregulated by resveratrol.

The transcription factors or proteins that affect gene expression were modulated with RES, LEC50 and LPS are listed in Table 5. RES alone downregulates gene expression of early growth response proteins 1 and 2 (*EGR1*, *EGR2*). RES and LPS + RES downregulated gene expression of several LPS induced transcription factors, and LEC50 alone upregulated gene expression of several transcription factors, such as, *NFKB1A*, *TRIM5*, *MAFF*, *HDAC9*, growth arrest and DNA damage-inducible protein 45 A (*GADD45A*), *IRF7*, etc. but downregulates *NOTCH3* and *OLIGO2*, like LPS. A total of 698 genes were upregulated when a combination of LPS + LEC50 was used, as shown in Figure 1. 

## 3. Discussion

This novel study provides strong evidence to support the conclusion that LPS (bacterial toxin from *E. coli*, normally present in the gut of some individuals) and LEC (plant toxin) showed a robust and synergistic reprograming of cells (XYZ/LMP to LMP) by upregulation of *PSMB8* (LMP7, CT-activity) and *PSMB9* (LMP2, post-acidic activity). A similar upregulation of inflammatory genes of mediators involved in signaling pathways in PBMCs from non-diabetic humans in ex vivo experiments was observed and LPS upregulated the IFN-γ and IL-4 induced genes. Conversely, LPS + RES, (a predominantly LMP7 proteasome inhibitor, encoded by *PSMB8* gene), downregulated expression of IFNγ- and IL-4 induced genes. These RNAseq data provide support for the hypothesis that RES and LEC can differentially affect the LPS-induced gene expression of proteasome subunits in ex vivo experiments and may also possess a potent immunomodulatory epigenetic effect on the host PBMCs. 

These results are significant, and they warrant further clinical research on reprograming the host’s immune system, depending on whether the person has a balanced immune system, is immuno-compromised, or is suffering from an autoimmune disorder. In addition, beneficial properties of RES in sepsis, heart disease and other inflammatory diseases must be studied in detail in humans [36,37]. In contrast, although the isoflavones in the soy products may be beneficial for heart disease, the lectins however, are unlikely to be effective for preventing heart disease [38], as we noticed that gene expression of endothelin-1 (*EDN1*), a potent vasoconstrictor, was upregulated and the insulin receptor (*INSR*) was downregulated by LPS or LEC. These may lead to other disorders such type 2 Diabetes Mellitus and arthritis. However, lectins may be beneficial to people with infections, such as tuberculosis or in certain cancers. 

The LPS-signaling pathway is well-established [39,40,41]. The main Toll-like receptor (TLR) genes upregulated by LPS in murine macrophages as analyzed by microarray technology included TLR4, TLR3, TLR1 and TLR2 [12]. The TLR4/MD-2 complex is the main receptor for LPS on the membrane surfaces [39,40,41,42]. LPS stimulation of TLR4/MD-2 leads to the activation of cells via adaptors such as MyD88 or TRIF-TRAM. However, many of the signaling mediators in Toll-receptor and other pathways are degraded by the UPS (Figure 3). MyD88 pathway ultimately leads to activation of the transcription factor NF-κB by upregulation of both p65 and p50, degradation of inhibitor IκB by the proteasome and the activation of NF-κB [41].

In addition to NF-κB, the activation of the TRIF/TRAM pathway after ligand-induced internalization into endosomes leads to the phosphorylation of IRF3, and induction of IFN-β [42,43,44,45]. However, in this study, when human PBMCs were activated with LPS or LEC, only gene expression of *p50* (but not *p65*, normally observed in murine cells) was upregulated. Moreover, gene expression of *TLR4*, *TLR2*, *TLR3 and TLR1* was not upregulated by LPS or LEC, because mRNA of these toll receptors was already present in the cell. In contrast, gene expression of *TLR7*, and *TLR9* (present in endosomes) was upregulated by LPS. Thus, LPS upregulated gene expression of *TRIF*, *MyD88*, *TLR7*, *TLR9*, *IRF7*, *TNF-α*, *IL-6*, *IL-12*, and *IL-2*; while LEC10 upregulated all these genes except for MyD88 (Figure 3), and LPS + RES downregulated TLR9 and TRIF. Proteasome subunits play a crucial role in the function of a cell. The effect of LPS, LEC50, and RES on gene expression of proteasome’s subunits X, Y, Z, LMP7, LMP2 is shown in Figure 4. LPS and LEC10/50 upregulated gene expression of LMP7 and LMP2, while LPS + RES inhibited it. RES alone repressed gene expression of LMP7 and it is also a proteasome inhibitor of LMP7 protease activity.

Resveratrol (3, 5, 4′-trihydroxystilbene) is a plant-based dietary compound found in grapes, berries, and red wine. Its beneficial effects have been attributed to its anti-inflammatory, antioxidant, anti-aging (due to increase in SIRT1), and anti-tumor properties [46,47,48,49,50,51,52]. We demonstrated earlier that pretreatment with RES inhibited inflammatory cytokines, TNF-α, IL-1β, IL-6, and nitric oxide levels, when added prior to LPS treatment of mouse Mφ and RES is a potent proteasome inhibitor [53,54,55]. However, no single mechanism has been presented. We now propose a novel probable mechanism for resveratrol. We have shown a direct effect of RES on the CT-like activity of the human monocyte’s proteasome containing LMP7, using the substrate peptide containing tryptophan in vitro. In this study, we found that RES downregulated gene expression of transcription factors such as early growth response proteins 1 and 2 (regulate differentiation) in PBMC. We have also demonstrated that most of the genes linked to dendritic-cell/T cell interaction and adaptive immune responses due to activation of LMP7, IL-4, and IFN-γ (also dependent on UPS), were robustly downregulated by RES. Therefore, the inhibition of the LMP7 subunit leads to a downregulation of both IFN-γ and IL-4 signaling pathways in response to LPS possibly via EGR which is also regulated by the proteasome. 

Lectins (LEC) are used as natural plant insecticides. When consumed by people, they trigger immune responses in people depending on the dose [56,57,58,59,60,61]. Lectins are present in abundant amounts in soybeans, red kidney beans, red hot peppers, potatoes, tomatoes, wheat germ, eggplant, nuts, raw unpasteurized milk, and mother’s breast milk. Lectins also induce IL-1β via the activation of inflammasomes [58]. Soybean lectin is a T-cell mitogen [56]. Most lectins are relatively heat-resistant to temperatures up to 70° C and rigorous heating conditions are required for their inactivation. Lectins present in soybeans, wheat grains (1 wheat grain has 10 μg of lectin), peppers, milk, and tomatoes are routinely consumed by people worldwide [56,57,58,59,60,61]. Our study has shown that LEC10 and LEC50 robustly induced gene expression of *PSMB8* (LMP7), and most inflammatory cytokines like LPS (a potent prototype and proinflammatory agonist, which induces both innate and acquired immunity) as described below (Figure 5A–D).

Gene expression of mediators involved in T-cell signaling pathway was also modulated with RES (anti-inflammatory) and LEC (inflammatory), since T cell signaling is also dependent on the LMP subunits of the proteasome [15]. LPS is usually not considered to be a direct T cell agonist, however, in the present study using human PBMCs, we observed that gene expression of *IFN-γ* (Th1), *IL-4* (Th2), *IL-2* (Th1), *IL-17 A* (Th17), *Treg* (IL-10, IL-20), and *PSMB8* (*LMP7)* was upregulated by LPS/LEC in a 3 h ex vivo treatment. Several transcription factor genes activated by *IFN-γ* (an amplifier of immune response), including *STAT2* and *STAT1* (transcription factors involved in JAK-STAT signaling), were also upregulated by LPS (Figure 5A–D).

Interestingly, RES downregulated gene expression *of IFN-γ*, *IFN-λ*, but not *IFN-β***.** RES inhibited LPS-induced *STAT1* more robustly than *STAT2*, thus inhibiting some of the IFN-γ induced genes (*IFN-γ*, *STAT1*, *TAP1*, *IFIT1*, *OAS1*, *IFITM1*, *MX1*, *GIP3*, *IRF9*, *IFI35 and GIP2*). We have also shown for the first time that LEC (with or without LPS) is an activator of the proteasome subunit *LMP7*, *IFN-λ*, *IFN-β*, *IFN-γ* and several other genes. Notably, LEC 10 and LEC 50 μg/mL upregulated expression of mostly the same genes as LPS 10 ng/mL. This study demonstrates the role of LEC and RES in upregulating/downregulating expression of *IL-4* and *IFN-γ* (Th2 and Th1 cytokines) and of mediators involved in signaling pathways in PBMC obtained from blood from non-diabetic controls.

Genes for several key enzymes involved in metabolic pathways linked to the proteasomes were also affected as explained above. Importantly, treatment of PBMC with LPS activates gene expression of proteins involved in several pathways, but LPS + RES leads to down-regulation of gene expression of some of the key enzymes involved in metabolism and other functions. These IFN-γ-induced genes include 2′-5′ oligoadenylate synthetase 1 (*OAS1)*, *OAS2* and *OAS3* (Figure 5A–D). These enzymes activate RNASE L leading to degradation of both viral and endogenous RNA. Ubiquitinated proteins are degraded by the proteasome. *USP18* gene encodes for an important enzyme that removes ISG-15 conjugates (ubiquitin-like molecules) from protein substrates (inhibits JAK-STAT signaling). It is also a type 1 interferon receptor repressor and belongs to a deubiquitinating protease family (DUB). Deficiency of this DUB is known to lead to upregulation of the immune system, and USP18 is degraded by the proteasome when produced in excessive amounts. *EIF2AK2* (protein kinase R) is an interferon-induced enzyme which catalyzes autophosphorylation of IκB and can induce cellular apoptosis to prevent viral spread. Collectively, LPS activates gene expression of proteins in UPS mentioned in this paragraph, RES inhibits these inflammatory proteins. Further protein analysis and clinical studies are needed to evaluate the potential therapeutic benefit of RES in reducing inflammation and impacting outcomes in human diseases, such as autoimmune conditions, diabetes, neurological problems, and cardiovascular diseases. 

LPS and LEC activated innate immune response via toll receptors and gene expression of proinflammatory cytokines such as *IL-1β*, *IL-6*, *TNF-α*, *CSF3*, *IFN-β*, *PSMB8*, and coagulation proteins *F3* and *F8.* Dendritic cells communicate with the T cells via the MHC-II molecules (with antigen) and T cell receptor TCR. There is also a crosstalk between the CD80 (dendritic cells) and CD28 proteins (T cells). In non-diabetic controls IL-12 is released from the macrophages/dendritic cells, resulting in differential induction of cytokines such as *IFN-γ*, *IL-2 (Th1)*, and *IL-4 (Th2)*, *IL-17 A (Th17) and IL-10 (Tregs)* as explained above (Figure 6) in response to LPS. RES repressed preferentially many of these genes associated with adaptive immune response via downregulation of *IFN-γ* and LMP7, in response to LPS (marked as RES-). In contrast, LEC alone upregulated expression of all these genes involved in innate and adaptive immune response. The mechanisms underlying these effects and the signaling pathways need to be further clarified.

Based on this novel study, we propose a simplified model that describes mechanisms for the beneficial and deleterious effects of dietary components in PBMCs The proteasome’s subunits can be linked to immune responses in the proposed model, LPS and IFN-γ (T cells) activates macrophages/monocytes and T cells by inducing LMP7 and LMP2 subunits of proteasomes and these are put into place within 3–24 h. LPS or LEC activates IL-12 in monocytes/macrophages, and this leads to the activation of Th1 cells via the crosstalk between T cell receptor (TCR) and MHC-II; and between CD28, (CTLA4) to release IFN-γ by Th1 (T) cells (Figure 7). The cells in early activated form **M1** possess XYZ subunits (in tissue cells or in mice) that are upregulated by LPS and new LMP2, 7 and 10 subunits are biosynthesized via activation of transcription factor, NF-κB by cleavage of IκB by the proteasomes. However, PBMC from controls were more differentiated in that subunit X is almost absent there was induction of LMP7 and LMP2 (but not LMP10), with LPS (active **M2**). In PBMC there was gene induction of p50/p50 subunits. Then, the gene expression of proteasome subunits is repressed or there is no further change (NC) in proteasome subunits with agonists during tolerance and the cells become refractory to LPS (**M3**). Proteasome subunits are degraded after ubiquitination, and this may lead to autophagy due to upregulation of Atg7 (an important autophagy gene induced by LPS) and recycling of cellular components [17]. Treatment with IFN-γ followed by LPS can overcome tolerance. We have previously shown that macrophages from LMP7/LMP10 knockouts do not induce gene and protein expression of IFN-γ, and nitric oxide in response to LPS suggesting that LMP subunits are crucial for expression of IFN-γ. However, treatment with IFN-γ can override this effect and the cells begin to induce nitric oxide [44,45]. 

LPS also causes tolerance after 8–24 h, and this may also be operative via the dietary compounds that affect gene encoding expression of *p50/p50* homodimers (cause tolerance), and *HDAC9* (deacetylase) is expected to cause epigenetic changes that shuts down new gene expression involved during adaptive immunity. This is due to DNA methylation and histone acetylation enzymes that are also regulated by the proteasome. Histone Acetyl Transferases (HATS) acetylate histones and activate the immune cells [62,63,64,65]. In contrast, histone deacetylases (HDACs) remove acetyl groups from histone tails making it harder for transcription factors to bind to DNA, leading to decreased levels of gene expression, and gene silencing or tolerance [63,64,65,66]. Our results suggest that *HDAC9* gene is the only deacetylase in human PBMCs that is induced by LPS or LEC, and repressed by RES. In addition, histone methylation [64] and Indoleamine 2,3-dioxygenase 2 (IDO2) (involved in tryptophan degradation and tolerance), known to be involved in the epigenetics and expression of these genes, was upregulated with LPS and LEC in PBMC [66,67,68].

We have established here that highly effective proteasome modulators, e.g., RES and LEC, commonly present in foods of human diets [46,47,48], can either down-regulate or up-regulate expression of these proteasome’s proteases in PBMCs, respectively. Differentiation of cells, Th1, Th2, and the pattern recognition pathways were differentially modulated by RES and LEC in the presence or absence of LPS. Importantly, most of the signaling pathways were upregulated with LPS and/or LEC; while the anti-inflammatory genes induced by *PPARγ* (for lipids and fatty acid synthesis); *LXR/RXR*, *VDR/RXR*, and *LXR/PPAR*. (These nuclear receptors regulate fatty acid and glucose homeostasis, VDR are receptors for steroids and vitamin D3), and the antioxidant action of vitamin C signaling pathways were downregulated by LPS and LEC (Figure 2). RES did not downregulate these pathways. RES downregulates gene expression of *PSMB8* (LMP7) alone. RES also downregulates crucial genes expressed in the LPS-induced Toll receptor and IFN-γ pathways. These results provide support for the hypothesis that dietary nutrients can robustly affect the LPS-induced signaling pathways in ex vivo experiments, however, more research is needed to confirm these important findings and determine which lectins have beneficial effects or adverse effects. 

## 4. Materials and Methods

### 4.1. Reagents

Deep rough chemotype LPS (Re-LPS) from *E. coli* D31m4 was purified as described by Qureshi et al. [69]. For tissue culture studies, RPMI 1640 Medium, heat-inactivated low-endotoxin fetal bovine serum (FBS), and gentamycin were purchased from Cambrex (Walkersville, MD, USA). RNeasy one-step kit was purchased from QIAGEN sciences (Germantown, MD, USA). Highly purified (Affinity column purified and stated by the company to have undetectable LPS, soybean lectin LEC from Glycine max) was purchased from Sigma-Aldrich (St. Louis, MO, USA). Trans-resveratrol (RES >99% HPLC pure) was also purchased from Sigma-Aldrich (St. Louis, MO, USA). PBMCs of normal healthy individual was purchased from Precision for Medicine (Frederick, MD, USA). Male healthy control was a 60-year-old, Caucasian (50 million cells). 

### 4.2. Detection of Cell Viability and Isolation of Total Cellular RNA

Viability and number of PBMCs were determined by trypan blue dye exclusion test and by using a cell counter. After treatment, cells were washed with PBS and total RNA was extracted by using RNeasy mini kit (Qiagen, Germantown, MD, USA) as per manufacturer’s instructions. 

### 4.3. Experiments Prior to RNAseq Analysis

Viable PBMCs (5 × 10^6^/10 mL/plate, duplicates) were treated either with 1. medium only, 2. RES (80 μM), 3. LPS (10 ng/mL), 4. LPS10 ng/mL+ RES 80 μM, 5. LEC10 (10 μg/mL), 6. LPS 10 ng/mL + LEC10 μg, 7. LEC 50 (50 μg/mL) and 8. LPS10 ng/mL + LEC50 μg/mL for 3 h. All samples were adjusted to contain the same final concentration of DMSO (0.2%). At 3 h of treatment, Total RNA was extracted from PBMCs and purified further using an affinity resin column (RNeasy, Qiagen, Chatsworth, CA, USA). 

### 4.4. Sample Preparation for RNAseq Analysis

Five to 8 μg of total RNA from each sample were provided to Novogene Global (CA, USA) for RNAseq analyses using the human Illumina system (Santa Clara, CA, USA). They assessed the purity of total RNA through the following tests, Nanodrop (OD 260/280), agarose gel electrophoresis for RNA purity, and Agilent 2100 analysis to check RNA integrity again. The following procedures were carried out by Novogene: mRNA enrichment, conversion to double-stranded cDNA, end repair, poly-A adaptor addition, fragment selection and PCR, library quality assessment and Illumina sequencing. An average of 44–61 million raw read counts were obtained. Novogene used the STAR software for alignment for RNA-seq data analysis. Read counts is proportional to gene expression levels, gene length and sequencing depth. The differential gene analysis was carried out on two samples (control vs. treatment group) at a time using the DESeq2 R package. The threshold of differential expression genes is log2 fold change >1, and *p* value < 0.05. 

### 4.5. Data Analysis and Network and Pathway Analysis

Gene expression data were first imported in the differentially expressed genes (DEG) program and numbers were corrected for differences in the IIlumina analysis. The numbers presented in Table 2, Table 3, Table 4 and Table 5 are averages from 2 incubations for each treatment; The log ratio values were normalized to a scale of 0 (instead of 1, which shows decimals) and expression values of upregulated genes showed positive numbers and the downregulated ones showed negative numbers (called normalized ratios, a log ratio of 2 is equivalent to a fold change of 4) and these ratios were imported into the Ingenuity Pathways Analysis (IPA) software (Ingenuity Systems, Mountain View, CA, USA) [12]. This is a web-based tool that is predicated on more than 200,000 full text articles and has information based upon 7900 human and mouse genes. This system categorizes genes into high level cellular functions and canonical pathways and has been used to characterize genes important in human systemic inflammation [12]. Genes found to be significantly activated were categorized based on different pathways and networks available in the database and ranked by score as described previously [12]. Genes identified as statistically different from background in terms of activation relative to control cells were analyzed and mapped into different pathways. 

## 5. Conclusions

Resveratrol may potentially be used for downregulation of chronic inflammation during autoimmune diseases, to inhibit the acquired immunity genes induced by LPS and other plant and viral toxins by downregulating gene expression of LMP7 (*PSMB8)*, *IFN-γ*. *and STAT1.* These results are consistent with our previous studies that showed that macrophages and T cells from LMP7/LMP10 double knockouts do not induce IFN-γ, STAT1, or nitric oxide (NO) with LPS [44}. Soybean lectin on the other hand, induces gene expression of mediators that is very similar to that of LPS, suggesting some degree of similarity in the signaling pathways by upregulating *PSMB8*, *STAT1*, and *IFN-γ with LEC*. Low doses of lectins may be beneficial to the immunosuppressed host, due to priming and enhancing IFN-γ and NO for killing bacteria and cancer cells. Lectins can also be used as adjuvants such as monophosphoryl lipid A (MPLA), which is used in vaccines and isolated from bacterial toxins, which was purified and characterized by us previously [70,71]. High doses of lectins may have deleterious effects, just like LPS. The knowledge gained from this study has shown that dietary nutrients (such as resveratrol and lectins) function to modulate gene expression, and if investigated further in clinical studies may lead to cures for heart disease, diabetes mellitus type 2, cancer, sepsis, MS, autoimmune diseases, viral infection, macular degeneration, psoriasis, Alzheimer’s, and other neurological disorders (depression) in both normal healthy and diseased individuals. Synthetic proteasome inhibitors have been extensively used for cancer and other inflammatory diseases, but these are very toxic as compared to natural products [72,73]. This information is therefore pivotal for future development of effective strategies for treatment of inflammatory diseases.

## Figures and Tables

**Figure 1 ijms-23-12946-f001:**
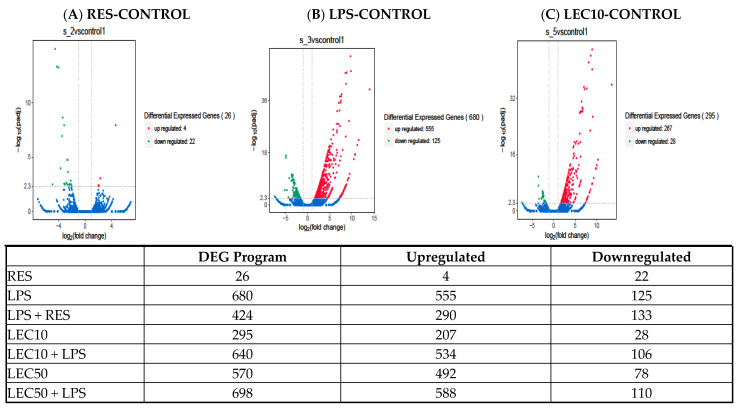
(**A**–**C**) The role of LPS, LEC10 and RES on gene expression of mediators involved in the agonist-induced signal transduction. Human PBMCs were treated with three compounds and the vehicle control for 3 h. RNA was extracted from the cells and analyzed using RNAseq. These data were obtained by using the DEG analysis. Volcano diagram depicts the differentially expressed genes (DEG) program data induced by LPS, LEC10 (red) and downregulated (green) by RES (-) and the table inset shows the number of DEG program data that are either upregulated or downregulated. This Log2 ratios DEG program data were downloaded into Ingenuity Pathways Analysis and Z score values (higher Z scores represent significant pathways) were obtained. Volcano diagram based on the DEG program data from PBMCs treated with RES, LPS or LEC10 (following table for Figure 1A–C).

**Figure 2 ijms-23-12946-f002:**
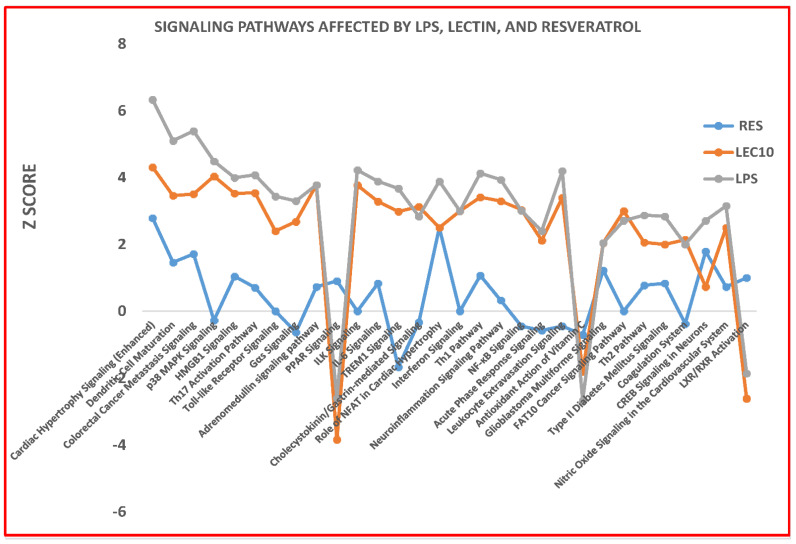
The role of LPS, LEC10 and RES on gene expression of mediators involved in the agonist-induced signaling pathways. The Log2 ratios of data obtained by the DEG program were downloaded into Ingenuity Pathways Analysis Program and Z score values (higher Z scores represent significant pathways) were obtained. Z scores against the signaling pathways were plotted.

**Figure 3 ijms-23-12946-f003:**
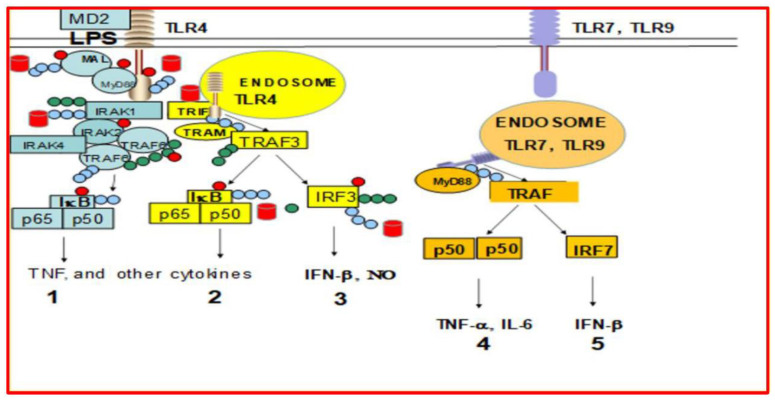
LPS-Signaling pathways. LPS signals via TLR4 (pathways 1–3). There are several mediators that are degraded by the proteasome such as MAL, IRAK1, TRAF6, IRF3, and IκB-α. The blue chains represent the K48-linked ubiquitins and those proteins are destined to be degraded by the proteasome [13]. The green chains in contrast, represent the K63-linked chains, which when linked to mediators have regulatory functions like TRAF6. The red circles represent phosphoryl groups, and the red barrels represent the proteasomes. Thus, proteasomes can regulate the degradation of signaling mediator proteins and can also activate transcription factors like NF-κB, IRF3, and IRF7 thus affecting gene expression of proteins. Thus, the proteasomes can regulate the degradation of proteins and synthesis of newly formed proteins.

**Figure 4 ijms-23-12946-f004:**
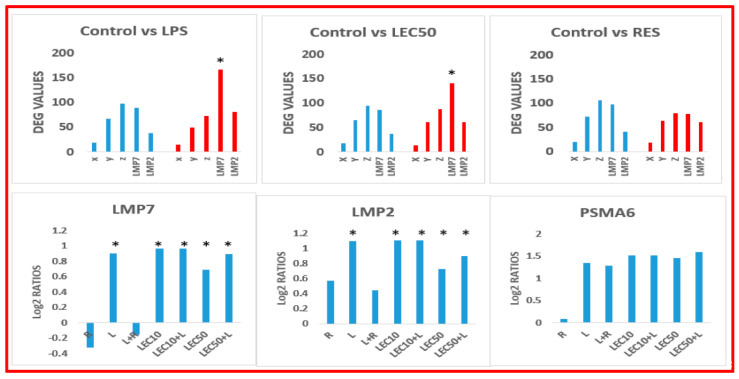
Functions of various components on gene expression of proteasome subunits. Human PBMCs were treated with three compounds and the vehicle control for 3 h as described in legend to Figure 1. These data were first extracted using the DEG analysis and plotted. The top row indicated the log2 ratios for control groups, and bottom row showed the log2 ratios for LMP7, LMP2 and PSMA6 (another proteasome subunit that was not downregulated by RES but was upregulated by LPS) with the different treatments (RES, LPS, RES + LPS, LEC10, LEC10 + LPS, LEC50 and LEC50 + LPS) were calculated relative to the control. *p* values were < 0.05. * significant gene expression.

**Figure 5 ijms-23-12946-f005:**
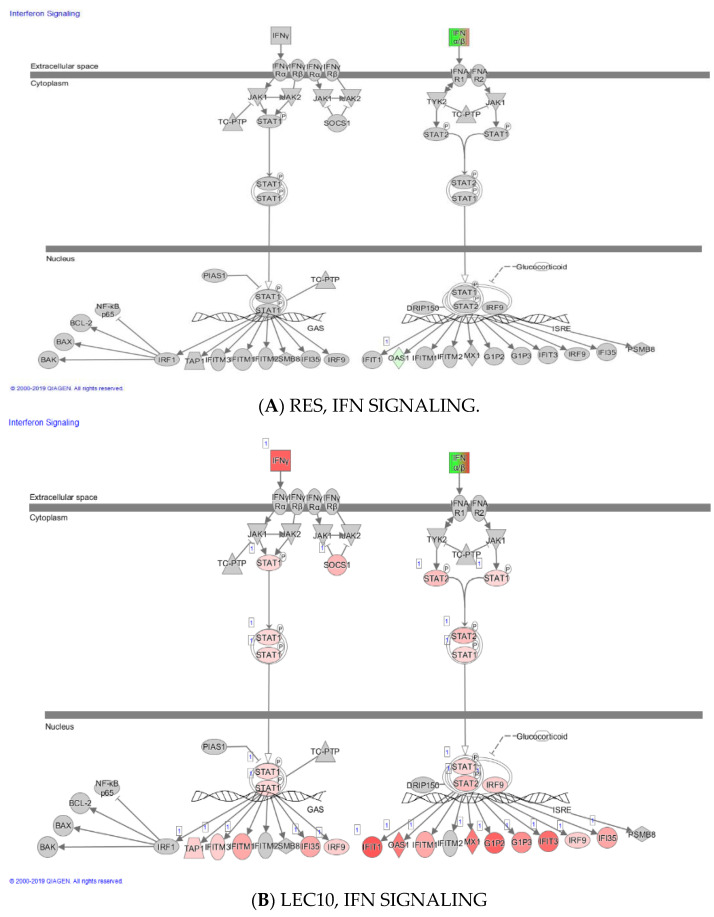
(**A**–**D**) The Interferon signaling pathway in human PBMCs. The interferon signaling pathway plays a critical role in the amplification of the immune response. The STAT1 and STAT2 transcription factors play a major role in the transcription of interferon activated genes. (**A**). RES, (**B**). LEC10, (**C**). LPS, and (**D**). LPS + RES. LPS and LEC activates gene expression of STAT1/STAT2, while RES downregulates it. Green, red, and grey colors denote inhibition, activation, and no change in gene expression, respectively.

**Figure 6 ijms-23-12946-f006:**
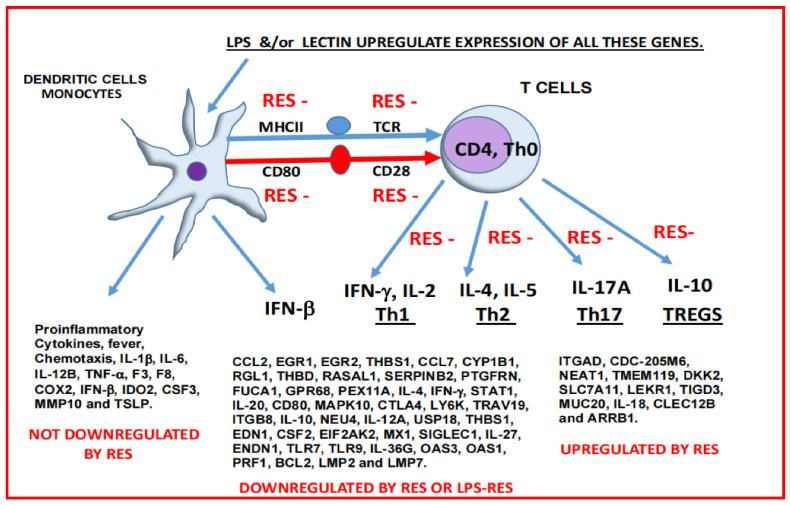
Summary of the effects of proteasome modulators on the immune system. LPS and LEC upregulates expression of several genes. RES either has no effect, downregulates, or upregulates expression of genes.

**Figure 7 ijms-23-12946-f007:**
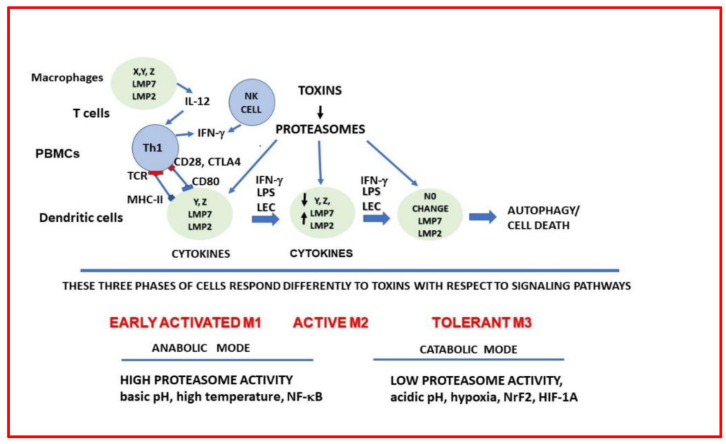
LPS, IFN-γ or LEC induces the reprogramming of the proteasomes with LMP7 and LMP2. The cell becomes tolerant and if the proteasome’s activities are completely inhibited then it may lead to autophagy. RES alone or in the presence can repress the induction of LMP7 and LMP2. RES can also inhibit the activity of the LMP7 protease more than the X protease.

**Table 1 ijms-23-12946-t001:** Genes downregulated with resveratrol alone *.

RES	Control	log2 Fold	* p * Value	Gene Name	Gene Description
4.635	107.400	−4.532	2.31 × 10^−20^	CCL2	chemokine_(C-C_motif)_ligand_2
4.0240	77.572	−4.266	1.96 × 10^−18^	EGR2	early_growth_response_2
136.81	2235.303	−4.030	3.57 × 10^−18^	THBS1	thrombospondin_1
10.798	111.972	−3.373	1.83 × 10^−13^	CCL7	chemokine_(C-C_motif)_ligand_7
121.84	1096.809	−3.170	1.19 × 10^−12^	CYP1B1	cytochrome_P450_family_1_subfamily_B_polypeptide_1
1.273	14.142	−3.466	1.68 × 10^−11^	CYP1B1-AS1	CYP1B1_antisense_RNA_1
5.755	37.271	−2.693	3.39 × 10^−9^	RGL1	ral_guanine_nucleotide_dissociation_stimulator-like_1
15.739	96.447	−2.614	3.48 × 10^−9^	THBD	thrombomodulin
2.699	16.375	−2.597	5.70 × 10^−8^	RASAL1	RAS_protein_activator_like_1_(GAP1_like)
64.894	292.372	−2.171	4.32 × 10^−7^	SERPINB2	serpin_peptidase_inhibitor_clade_B_(ovalbumin)_member2
0.764	5.423	−2.816	7.48 × 10^−7^	PTGFRN	prostaglandin_F2_receptor_inhibitor
42.635	182.580	−2.098	1.07 × 10^−6^	FUCA1	fucosidase_alpha-L-_1_tissue
10.085	44.555	−2.142	1.22 × 10^−6^	GPR68	G_protein-coupled_receptor_68
0.050	1.754	−4.938	1.30 × 10^−06^	PEX11A	peroxisomal_biogenesis_factor_11_alpha
8.251	36.420	−2.141	1.36 × 10^−6^	EGR1	early_growth_response_1
1.782	9.410	−2.395	1.36 × 10^−6^	LY6K	lymphocyte_antigen_6_complex_locus_K
0.458	3.828	−3.044	1.39 × 10^−6^	TRAV19	T_cell_receptor_alpha_variable_19
2.597	12.813	−2.299	1.58 × 10^−6^	ITGB8	integrin_beta_8
0.865	5.582	−2.679	1.64 × 10^−6^	IL10	interleukin_10
13.498	55.933	−2.050	2.61 × 10^−6^	NEU4	sialidase_4

* These data represent the genes that are downregulated by resveratrol alone as compared to a vehicle control in PBMCs incubated for 3 h. RNA was extracted and subjected to RNAseq analysis. This data has been obtained by using the DEG program.

**Table 2 ijms-23-12946-t002:** LPS-induced genes downregulated by resveratrol (RES) *.

LPS + RES	LPS	log2 Fold	* p * Value	Gene Name	Gene Description
0.0529	9.74207	−7.3514	1.41 × 10^−18^	CCL8	chemokine_(C-C_motif)_ligand_8
27.447	476.382	−4.1171	1.49 × 10^−18^	CYP1B1	cytochrome_P450_family_1_subfamily_B_polypeptide_1
7.7361	135.790	−4.1325	4.13 × 10^−18^	CCL2	chemokine_(C-C_motif)_ligand_2
125.420	1442.97	−3.5241	7.36 × 10^−15^	RSAD2	radical_S-adenosyl_methionine_domain_containing_2
4.715	61.064	−3.6929	8.29 × 10^−15^	RGL1	ral_guanine_nucleotide_dissociation_stimulator-like_1
8.954	108.360	−3.596	1.04 × 10^−14^	CCL7	chemokine_(C-C_motif)_ligand_7
62.895	689.129	−3.453	2.18 × 10^−14^	OAS3	2′-5′-oligoadenylate_synthetase_3_100kDa
0.105	6.422	−5.834	5.39 × 10^−14^	IL19	interleukin_19
57.067	593.613	−3.378	6.55 × 10^−14^	CMPK2	cytidine_monophosphate_(UMP-CMP) kinase_2_mitochondrial
5.0337	54.642	−3.438	2.76 × 10^−13^	CXCL11	chemokine_(C-X-C_motif)_ligand_11
6.358	63.133	−3.310	9.45 × 10^−13^	IFNG	interferon_gamma
2.331	25.960	−3.473	1.33 × 10^−12^	SIGLEC1	sialic_acid_binding_Ig-like_lectin_1_sialoadhesin
0.688	11.102	−3.997	1.61 × 10^−12^	GGT5	gamma-glutamyltransferase_5
23.526	194.569	−3.047	1.01 × 10^−11^	NEU4	sialidase_4
0.635	9.252	−3.849	2.12 × 10^−11^	IL10	interleukin_10
1.006	12.136	−3.582	2.65 × 10^−11^	EDN1	endothelin_1
6.093	50.397	−3.046	3.61 × 10^−11^	CSF2	colony_stimulating_factor_2_(granulocyte-macrophage)
17.538	134.484	−2.938	5.35 × 10^−11^	USP18	ubiquitin_specific_peptidase_18
5.351	43.920	−3.035	5.93 × 10^−11^	EGR2	early_growth_response_2
0.105	3.973	−5.142	2.01 × 10^−10^	LAMA2	laminin_alpha_2
1023.76	6937.936	−2.760	2.90 × 10^−10^	THBS1	thrombospondin_1
171.677	1129.917	−2.718	5.22 × 10^−10^	MX1	myxovirus_(influenza_virus)_resistance_1_interferon-inducible_protein_p78_(mouse)
237.487	1548.772	−2.705	6.12 × 10^−10^	HELZ2	helicase_with_zinc_finger_2_transcriptional_coactivator
68.035	395.343	−2.538	5.47 × 10^−9^	EIF2AK2	eukaryotic_translation_initiation_factor_2-alpha_kinase_2
0.317	3.156	−3.284	1.46 × 10^−6^	GAPDHP14	glyceraldehyde-3-phosphate_dehydrogenase_pseudogene_14
237.328	983.024	−2.050	1.54 × 10^−6^	OAS2	2′-5′-oligoadenylate_synthetase_2_69/71kDa
4.821	21.606	−2.162	2.29 × 10^−6^	DAGLA	diacylglycerol_lipase_alpha

* These data represent the genes that are modulated (upregulated/downregulated) by RES + LPS as compared to LPS alone in PBMCs, incubated for 3h. RNA was extracted and subjected to RNAseq analysis. This data has been obtained by using the DEG program.

**Table 3 ijms-23-12946-t003:** (**A**) Genes upregulated by LEC 10 alone *. (**B**) Genes not upregulated by LEC10 to the same extent as LPS *. * These data represent the genes that are upregulated by LEC10 alone as compared to a LPS control in PBMCs, incubated for 3 h. RNA was extracted and subjected to RNAseq analysis. This data has been obtained by using the DEG program. (**C**) Genes that are upregulated by LPS, LEC10, LEC10 + LPS, LEC50 and LEC50 + LPS *.

**(A)**								
**LEC10**	**Control1**	**log2** **Fold**	***p* Value**	**Gene Name**	**Gene Description**			
1105.120	2.180	8.980	1.96 × 10^−51^	IL6	interleukin_6_(interferon_beta_2)			
1116.645	2.824	8.623	2.62 × 10^−49^	IL1A	interleukin_1_alpha			
2159.807	7.532	8.162	3.34 × 10^−47^	CCL4	chemokine_(C-C_motif)_ligand_4			
226.943	0.446	8.969	5.46 × 10^−45^	CCL3L1	chemokine_(C-C_motif)_ligand_3-like_1			
77.794	0	13.49	1.22 × 10^−40^	CSF3	colony_stimulating_factor_3_(granulocyte)			
1,9385.16	146.59	7.046	8.24 × 10^−40^	IL1B	interleukin_1_beta			
199.06	0.941	7.713	2.26 × 10^−39^	CCL20	chemokine_(C-C_motif)_ligand_20			
488.089	3.270	7.218	3.70 × 10^−39^	F3	coagulation_factor_III_(thromboplastin_tissue_factor)			
2886.299	31.916	6.498	1.50 × 10^−35^	PTGS2, COX2	Prostaglandin-endoperoxidesynthase_2_ (prostaglandin_ G/H_synthase_and_cyclooxygenase)			
178.034	1.784	6.635	1.10 × 10^−33^	IRG1	immunoresponsive_1_homolog_(mouse)			
128.475	1.139	6.808	2.52 × 10^−33^	CCL3	chemokine_(C-C_motif)_ligand_3			
26.226	0.346	6.212	9.88 × 10^−24^	CSF2	colony_stimulating_factor_2 (granulocyte-macrophage)			
439.770	15.065	4.866	1.58 × 10^−23^	TNF	tumor_necrosis_factor			
2759.83	109.77	4.651	1.99 × 10^−22^	CXCL2	chemokine_(C-X-C_motif)_ligand_2			
2523.37	105.21	4.583	5.92 × 10^−22^	CXCL3	chemokine_(C-X-C_motif)_ligand_3			
8.210	0.198	5.325	4.05 × 10^−15^	IL36G	interleukin_36_gamma			
3,0181.18	2646.306	3.511	8.38 × 10^−15^	IL8	interleukin_8			
659.862	74.437	3.147	1.68 × 10^−12^	G0S2	G0/G1switch_2			
758.803	87.323	3.119	2.47 × 10^−12^	NLRP3	NLR_family_pyrin_domain_containing_3			
3.248	0.198	3.989	3.42 × 10^−8^	GCKR	glucokinase_(hexokinase_4)_regulator			
17.130	4.113	2.056	8.44 × 10^−6^	HDAC9	histone_deacetylase_9			
8.033	1.685	2.248	8.51 × 10^−6^	GGT5	gamma-glutamyltransferase_5			
15.948	3.964	2.006	1.41 × 10^−5^	RBKS	ribokinase			
2.421	0.297	2.997	1.87 × 10^−5^	GPR141	G_protein-coupled_receptor_141			
**(B)**								
**LPS**	**LEC10**	**log2** **Fold**	***p* Value**	**Gene Name**	**Gene Description**			
66.397	6.369	3.380	3.99 × 10^−13^	IFNG	Interferon-gamma			
131.870	16.719	2.979	3.19 × 10^−11^	USP18	ubiquitin_specific_peptidase_18			
44.223	6.687	2.724	2.05 × 10^−9^	CXCL11	chemokine_(C-X-C_motif)_ligand_11			
1395.944	265.063	2.396	2.88 × 10^−8^	RSAD2	radical_S-adenosyl_methionine_domain_containing_2			
393.764	74.572	2.400	2.95 × 10^−8^	IFI6	interferon_alpha-inducible_protein_6			
15.275	2.653	2.521	1.50 × 10^−7^	CACNA1A	calcium_channel_voltage-dependent_P/Q_type_alpha_1A_subunit			
7.329	1.114	2.709	3.69 × 10^−7^	FAM19A2	family_with_sequence_similarity_19_(chemokine_(C-C_motif)-like)_member_A2			
91.465	19.956	2.196	4.74 × 10^−7^	LAMP3	lysosomal-associated_membrane_protein_3			
20.017	3.980	2.328	5.25 × 10^−7^	BCL2L14	BCL2-like_14_(apoptosis_facilitator)			
24.575	5.095	2.268	7.22 × 10^−7^	SIGLEC1	sialic_acid_binding_Ig-like_lectin_1_sialoadhesin			
1500.467	345.580	2.118	7.24 × 10^−7^	ISG15	ISG15_ubiquitin-like_modifier			
3.202	0.318	3.301	9.76 × 10^−7^	CCNA1	cyclin_A1			
**(C)**								
** Symbol **	** Entrez Gene Name **			** LPS **	** LEC10 **	** LEC10 + LPS **	** LEC50 **	** LEC50 + LPS **
IL6	interleukin 6			9.67	9.724	9.724	9.917	9.865
IL1A	interleukin 1 alpha			8.788	8.816	8.816	9.167	8.843
CCL20	C-C motif chemokine ligand 20			8.674	8.764	8.764	9.231	8.888
CSF3	colony stimulating factor 3			14.018	14.054	14.054	14.629	14.428
INHBA	inhibin subunit beta A			7.711	7.668	7.668	7.583	7.587
F3	coagulation factor III, tissue factor			7.579	7.674	7.674	8.117	7.857
IL1B	interleukin 1 beta			7.304	7.380	7.380	7.749	7.572
ACOD1	aconitate decarboxylase 1			7.546	7.539	7.539	7.252	7.811
TNFAIP6	TNF alpha induced protein 6			6.695	6.671	6.671	6.757	6.659
CSF2	colony stimulating factor 2			7.311	7.399	7.399	8.025	7.84
CXCL11	C-X-C motif chemokine ligand 11			6.662	6.324	6.324	5.338	6.432
IL12B	interleukin 12B			9.513	9.761	9.761	9.497	9.525
IDO2	indoleamine 2,3-dioxygenase 2			8.616	9.145	9.145	8.351	9.084
IL36RN	interleukin 36 receptor antagonist			8.515	7.942	7.942	8.142	8.774
IFNB1	interferon beta 1			8.443	8.194	8.194	8.534	8.851

* These data represent the genes that are upregulated by LEC10 alone as compared to a vehicle control in PBMCs, incubated for 3h. RNA was extracted and subjected to RNAseq analysis. This data has been obtained by using the DEG program. These data represent the genes that are upregulated by LPS, LEC10, LEC10 + LPS, LEC50 and LEC50 + LPS, as compared to vehicle controls in PBMC, incubated for 3h. RNA was extracted and subjected to RNAseq analysis. This data has been obtained by using the DEG program. The log2 ratios were imported into the Ingenuity Pathways Analysis.

**Table 4 ijms-23-12946-t004:** Important genes in LPS induced pathways that are modulated by RES and LEC.

Symbol	RES	LPS	RES + LPS	LEC10	Gene Description
IFNG	0.190	4.913	1.586	4.953	Interferon γ
IFN-l	Nc	6.491	-	9.194	Interferon lamda
IFNB1	Nc	8.443	7.453	8.194	Interferon β
TNF-α	−0.043	5.790	5.392	5.986	Tumor necrosis factor
IL12A	2.130	6.216	4.646	6.080	Interleukin 12 α
IL12B	Nc	9.513	8.337	9.761	Interleukin 12 β
IL27	−0.636	3896	0.963	4.189	Interleukin 27
IL36RN	Nc	8.515	4.204	7.942	Interleukin 36 receptor antagonist
IL-36B	Nc	5.984	3.280	5.846	Interleukin 36 β
IL-20	Nc	5.984	5.164	6.833	Interleukin 20
IL6	−0.095	9.670	10.139	9.724	Interleukin 6
IL1B	−0.956	7.304	7.467	7.38	Interleukin 1 β
IL1R1	−0.904	1.184	0.095	1.170	Interleukin 1 receptor 1
TGFB2	1.636	3.349	1.609	3.377	Transforming growth factor-β2
MYD88	−0.249	1.050	−0.134	0.975	Myeloid differentiation primary response 88
RIPK1	0.128	1.031	0.485	1.012	Receptor-interacting serine/threonine-protein kinase 1
NLRP3	0.361	3.019	4.083	3.124	NLR family pyrin domain containing 3 (NLRP3)
INSR	−0.424	−1.683	−1.424	−2.114	Insulin receptor
INSM1	Nc	6.173	3.28	6.064	Insulinoma-associated protein 1
F2RL1	0.175	1.867	1.32	1.842	Thrombin receptor like 1,3
LDLR	0.300	1.095	0.406	1.105	Lipoprotein receptor
CXCL10	0.873	4.645	2.060	4.560	C-X-C motif chemokine ligand 10
CXCL8	−0.893	3.913	3.845	3.974	chemokine (C-X-C motif) ligand 8
CD80	0.936	3.12	1.541	3.183	Cluster of differentiation 80
CD40	0.383	1.284	1.495	1.188	Cluster of differentiation 40
CTLA4	−0.119	1.171	0.451	1.181	Cytotoxic T-lymphocyte-associated protein 4
STAT1	0.090	1.356	−0.206	1.249	Signal transducer and activator of transcription 1
HLA-DQB	1.360	3.122	2.148	3.206	HLA class II DQB
IRF7	0.178	2.492	1.519	2.556	Interferon regulatory factor 7
TLR7	−0.116	3.156	1.854	3.089	Toll like receptor 7
NCOA7	−0.477	2.355	0.015	2.330	Nuclear receptor coactivator
PTGER3	−1.025	2.269	0.233	2.560	Prostaglandin EP3 receptor
SIGLEC1	0.490	3.694	0.204	3.582	Sialo adhesin

**Table 5 ijms-23-12946-t005:** Transcription factor genes modulated by different treatments *.

2 RES	3LPS	4LPS + RES	5LEC10	6LEC10 + LPS	7LEC50	8LEC50 + LPS
*EGR1-*	*HDAC9*	*NFKBIA*	*HDAC9*	*HDAC9*	*HDAC9*	*HDAC9*
*EGR2-*	*PARP12*	*GADD45A*	*NFKBIA*	*PARP12*	*NOTCH3-*	*PARP12*
	*NOTCH3-*	*EGR1*	*GADD45A*	*NOTCH3-*	*NFKBIA*	*NOTCH3-*
	*NFKBIA*	*EGR2-*	*EHF*	*NFKBIA*	*NCOA7*	*NFKBIA*
	*KLF5*	*IRF5-*	*ETS2*	*NCOA7*	*GADD45A*	*KLF5*
	*NCOA7*	*EHF*	*E2F7*	*GADD45A*	*EGR1*	*NCOA7*
	*GADD45A*	*ETS2*	*MAFF*	*TRIM25*	*TRIM5*	*GADD45A*
	*TRIM25*	*E2F7*		*TRIM5*	*EHF*	*EGR1*
	*TRIM5*	*ZNF668-*		*EHF*	*PML*	*TRIM25*
	*EHF*	*HIC1*		*PML*	*ETS2*	*TRIM5*
	*PML*	*OLIG2-*		*ETS2*	*E2F7*	*TBX3-*
	*ETS2*			*E2F7*	*MAFF*	*EHF*
	*E2F7*			*MAFF*	*IRF7*	*PML*
	*MAFF*			*IRF7*	*OLIG2-*	*ETS2*
	*IRF7*			*OLIG2-*		*E2F7*
	*OLIG2-*					*MAFF*
						*IRF7*
						*OLIG2-*

* The transcription factor genes that were upregulated in samples. The transcription factors highlighted in yellow were downregulated, while the rest were upregulated. *EGR1* and *EGR2* = Early growth response proteins 1 and 2, *HDAC9* = Histone deacetylase 9, *PARP12* = Poly [ADP-ribose] polymerase 12, *NOTCH3* = Neurogenic locus notch homolog protein 3, *NFKBIA* = IκBα (nuclear factor of kappa light polypeptide gene enhancer in B-cells inhibitor, alpha), *KLF5* = Krueppel-like factor 5, *NCOA7* = Nuclear receptor coactivator 7, *GADD45A* = Growth arrest and DNA-damage-inducible protein 45 alpha,*TRIM25* = Tripartite motif-containing protein 25, *TRIM5* Tripartite motif-containing protein 5, *EHF* = ETS homologous factor, *PML*, Transcription factor, *ETS2* = Protein C-ETS2, *E2F7* = Transcription factor, *MAFF* = v-maf avian musculoaponeurotic fibrosarcoma oncogene homolog F, *IRF7* = Interferon regulatory factor 7, *TBX2* = T-box transcription factor 2, *OLIG2* = Oligodendrocyte transcription factor 2, and *HIC1* = Hypermethylated in cancer 1 protein.

## Data Availability

All data generated or analyzed during this study are included in this article.

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
