# Peer review of "Reprograming of Gene Expression of Key Inflammatory Signaling Pathways in Human Peripheral Blood Mononuclear Cells by Soybean Lectin and Resveratrol"

_ijms, 2022, doi:10.3390/ijms232112946_

Round 1

Reviewer 1 Report

In this study, the authors aimed to observe the gene expression changes in humans PBMCs, treated with resveratrol and soybean lectin with or without LPS. Although the subject may be of interest, the manuscript is written in a very confusing manner and is not focused. The data are presented in a very confusing manner as well, almost impossible to follow with what the authors are trying to describe.

1) Throughout the manuscript, the authors use words like regulate, modulate, which is not very clear. The data from previous reports should state clearly whether the changes were beneficial or deleterious, upregulated or downregulated.

2) Line 45: wrong reference format

3) Line 55: why use italic?

4) Figure 1: there is no A and B annotation

5) Tables: it is not appropriate to put all upregulated/downregulated genes in the main text. Most should be removed to supplementary, only leaving the top priority genes may be.

6) Line 111, 112: what is the criteria for describing maximal and some?

7) Rather than describing RES, LPS+RES, etc., the authors should first collectively describe the effect of LPS and how combination of RES and LEC10 changes the effect of LPS alone. What is the aim for describing RES and LEC alone?

8) Line 143: 'To facilitate the analysis' - I am not sure how Table 1A and 1B facilitates the analysis.

9) Table 1. A -> Table 1A.

10) The descriptions in results does not seem to be focusing on UPS but overall. Thus, the manuscript loses its focus.

Author Response

Reviewer #1.

  1. The term “modulates” or “regulate” is used for upregulation and downregulation. We have rewritten the manuscript based on this comment. RES downregulates gene expression, while LEC upregulates inflammatory genes, but when we mention both, we write the word “modulate”.
  2. We changed the reference format on line 45.
  3. We removed the italics.
  4. We removed Fig 1B. There were too many figures.
  5. Tables 1, 2 and 3 have been moved to the supplementary file. We have mentioned only a few genes in the text.
  6. We changed that the word “maximal”.
  7. We have now described the effects of LPS first and then RES and LEC. However, we also used the dietary components alone. For instance, LEC alone is upregulating all genes like LPS. RES can downregulate some genes alone. In the genetically modified soybean (GMO) varieties the lectins in soybeans have doubled. This may cause deleterious effects after consuming soybeans.
  8. We changed the word facilitate.
  9. Table 1. A has been changed to Table 1A.

We thank the reviewers for doing a great job on reviewing this manuscript.  We hope that the revised version of this manuscript is suitable for publication in your journal.

Reviewer 2 Report

The manuscript entitled “Reprograming of Gene Expression of Key Inflammatory Signaling Pathways in Human Peripheral Blood Mononuclear Cells by Soybean Lectin and Resveratrol “by Qureshi et al is an impressive manuscript evaluating the gene expression changes in PBMCs with treatment of LPS, Lectin, Resveratrol and combinations of the three. Evaluation of the gene expression changes are quite comprehensive, but unfortunately that is also one the largest issues with digestibility/understandability of the data.

Some of the bigger issues are as follows:

- Much of the tables (specifically Tables 2A-C and 3A-D) could be moved to supplemental materials.

- There are also some editing choices that were not clearly explained specifically the bolding of some of the groups throughout the tables.

- Table 4 referenced in line 249 is not present in the manuscript.

- Table 3C lacks a title.

- Figure 2 many of the group names have been cut off and the data overlaps the writing making it further harder to read.

- Figure 6A is duplicated.

- Figures 5 and 6 are a bit unwieldly to interpret, if there is a possible way to combine the groups to summarize the findings, it would make the point much clearer.

Minor edits:

-       Table 6 could benefit from a more intuitive labeling of upregulated vs downregulated (green vs red, arrows up and down, etc).

Overall the data presented in the manuscripts is of interest. It provides a greater understanding of how these molecules alter gene expression in PBMCs, particularly in the pathways the authors pointed out. Figure 8 is a wonderful figure that distills the information into a relatable figure that, if the data tables were to be moved to supplements, readers could appreciate the findings and look deeper at the data if needed.  

Author Response

Reviewer #2.

  1. Tables 1, 2 and 3 have been moved to the supplementary file.
  2. We had bolded all the names of enzymes, but we have removed the bolding now.
  3. We have checked all the references.
  4. The title for Table 3C. has been added.
  5. We have corrected Figure 2.
  6. Fig 6A. has been deleted.
  7. Figs 5. has been deleted and another new figure on the LPS signaling pathway has been added. Fig. 6 on the coagulation pathway has also been deleted. We are now discussing only the LPS signaling pathway and the IFN-g + log2 values denote upregulation, while -log2 fold values denote downregulation.
  8. We have Fig 8. in the manuscript. Thank you for the comment. Now the labeling of the Figures has changed, and this is a more focused manuscript.

We thank the reviewers for doing a great job on reviewing this manuscript.  We hope that the revised version of this manuscript is suitable for publication in your journal.

Round 2

Reviewer 1 Report

- The authors' response should include the original comment by the reviewers. My comments are not visible in the document that the authors have submitted and hence don't know what the authors are replying to.

- The revised manuscript should not use track changes but highlight with RED where the changes are made. It is difficult to read in this format throughout the manuscript.

- In the current form, I do not think the comments made by the reviewers are sufficiently answered during the revision process.

Author Response

Author's Reply to the Review Report (Reviewer 1; First Round).

Comments and Suggestions for Authors. In this study, the authors aimed to observe the gene expression changes in humans PBMCs, treated with resveratrol and soybean lectin with or without LPS. Although the subject may be of interest, the manuscript is written in a very confusing manner and is not focused. The data are presented in a very confusing manner as well, almost impossible to follow with what the authors are trying to describe:

1) Throughout the manuscript, the authors use words like regulate, modulate, which is not very clear. The data from previous reports should state clearly whether the changes were beneficial or deleterious, upregulated or downregulated.

The term “modulates” or “regulate” is used for upregulation and downregulation. We have rewritten the whole manuscript based on this comment. Resveratrol (RES) downregulates gene expression, while Soybeans Lectin (LEC) upregulates inflammatory genes, but when we mention both regulations (downregulation & upregulation), we write the word “modulate”. In Abbreviation Page-20, Line 598.

2) Line 45: wrong reference format

The reference format on line 45 has been changed. Page 6, Line 46.

3) Line 55: why use italic?

The italics have been removed. Page 2, Line 59

4) Figure 1: there is no A and B annotation.

The annotation in Figure 1 has been added 1A,1B. Page 3, Line 128.

5) Tables: it is not appropriate to put all upregulated/downregulated genes in the main text. Most should be removed to supplementary, only leaving the top priority genes may be.

Tables 1, 2 and 3 have been moved to the supplementary file. We have mentioned only a few genes in the text. Page 6, Line 178; Page 7, line 193; Page 8, Line 210.

6) Line 111, 112: what is the criteria for describing maximal and some?

The word “maximal” has been deleted. “698 genes” has been added on Page 3, Line 117.

7) Rather than describing RES, LPS + RES, etc., the authors should first collectively describe the effect of LPS and how combination of RES and LEC10 changes the effect of LPS alone. What is the aim for describing RES and LEC alone?

The effects of LPS first and then RES and LEC have been described as suggested by the reviewer. However, we also used the dietary components alone. For instance, LEC alone is upregulating all genes like LPS. RES can downregulate some genes alone as suggested by reviewer # 1. In the genetically modified soybeans (GMO) varieties, the lectins in soybeans have doubled. This may cause deleterious effects after consuming soybeans.

8) Line 143: 'To facilitate the analysis' - I am not sure how Table 1A and 1B facilitates the analysis.

The word “facilitate” has been changed with “to analyze” (Page 5, Line 147).

9) Table 1. A -> Table 1A.

Table 1. A has been changed to Table 1A as pointed out by the reviewer. Page 5, Line 149.

10) The descriptions in results does not seem to be focusing on UPS but overall. Thus, the manuscript loses its focus.

The results and the discussion sections have been rewritten according to reviewer’s suggestion. Starting from Page 3, Line 114.

Author's Reply to the Review Report (Reviewer # 1; Second Round).

  1. The authors' response should include the original comment by the reviewers. My comments are not visible in the document that the authors have submitted and hence don't know what the authors are replying to.

The reply has been modified accordingly.

  1. The revised manuscript should not use track changes but highlight with RED where the changes are made. It is difficult to read in this format throughout the manuscript.

All the changes have been highlighted with red.

  1. In the current form, I do not think the comments made by the reviewers are sufficiently answered during the revision process.

The manuscript has been revised very carefully to the best our knowledge. It is very short (4780 KB to 3061 KB) and focused as mentioned earlier. The manuscript has been modified further to accommodate reviewer # 1 concerns. Page 18, Lines 504-507, 509.   

Reviewer 2 Report

The authors have done a great job at editing the manuscript to highlight the significance and digestibility of the data. They have addressed each of the comments and no concerns remain about the acceptance of the manuscript for publication. Great job!

Author Response

Author's Reply to the Review Report (Reviewer # 2; First Round).

  1. Much of the tables (specifically Tables 2A-C and 3A-D) could be moved to supplemental materials.

Tables 1, 2 and 3 have been moved to the supplementary file. Page 24, Line 809.

  1. There are also some editing choices that were not clearly explained specifically the bolding of some of the groups throughout the tables.

The bolded names of some of groups (all enzymes) in tables have been removed.

  1. Table 4 referenced in line 249 is not present in the manuscript.

Table 4 is on Page 9, Line 217.

  1. Table 3C lacks a title.

The title for Table 3C has been added. Page 8, Line 210.

  1. Figure 2 many of the group names have been cut off and the data overlaps the writing making it further harder to read.

The Figure 2 has been corrected. Page 5, Line 162.

  1. Figure 6A is duplicated.

Duplicated Figure 6A has been deleted.

  1. Figures 5 and 6 are a bit unwieldly to interpret, if there is a possible way to combine the groups to summarize the findings, it would make the point much clearer.

Original Figures 5A-D have been deleted and another new figure on the LPS signaling pathway has been added Figure 3 (pg. 3) on the LPS-Signaling pathway. We are now discussing only the LPS signaling pathway (Figure 3) and the IFN-g pathway (5A-D, pgs. 14 and 15). The + log2 values denote upregulation, while -log2 fold values denote downregulation.

Reviewer # 2 (Second Round).

The authors have done a great job at editing the manuscript to highlight the significance and digestibility of the data. They have addressed each of the comments and no concerns remain about the acceptance of the manuscript for publication. Great job!

Thank you for the comments. This is a more focused manuscript. We thank the reviewers for doing a great job on reviewing this manuscript.  We hope that the revised version of this manuscript is suitable for publication in your journal.

Round 3

Reviewer 1 Report

The authors have replied to the comments made in full.